# On Bed Form Resistance and Bed Load Transport in Vegetated Channels

**Jennifer G. Duan * and Khalid Al-Asadi**

Department of Civil and Architectural Engineering and Mechanics, University of Arizona, 1209 E. 2nd Street, Tucson, AZ 85721, USA
* Correspondence: gduan@arizona.edu

**Abstract:** A set of laboratory experiments were conducted to study the impact of vegetation on bed form resistance and bed load transport in a mobile bed channel. Vegetation stems were simulated by using arrays of emergent polyvinyl chloride (PVC) rods in several staggered configurations. The total flow resistance was divided into bed, sidewall, and vegetation resistances. Bed resistance was further separated into grain and bed form (i.e., ripples and dunes) resistances. By analyzing experimental data using the downhill simplex method (DSM), we derived new empirical relations for predicting bed form resistance and the bed load transport rate in a vegetated channel. Bed form resistance increases with vegetation concentration, and the bed load transport rate reduces with vegetation concentration. However, these conclusions are obtained by using experimental data from this study as well as others available in the literature for a vegetated channel at low concentration.

**Keywords:** bed form resistance; bed load transport; vegetated channel; bed form; vegetation concentration; downhill simplex method

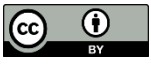

## 1. Introduction

Flow in vegetated rivers is characterized by the interaction between flow, channel boundary, and vegetation canopy. Vegetation characteristics, such as vegetation type, distribution, density, flexibility, and degree of submergence, all affect flow depth, velocity, hydraulic radius, and energy slope. The drag force exerted by vegetation increases flow resistance [1,2]. Because of this drag, flow through vegetated reaches decelerates, and the velocity is smaller than that in non-vegetated reaches [3,4]. Vegetation also changes near-bed turbulent characteristics and therefore affects sediment transport and scour formation around vegetation stems [5–8].

Resistance in a vegetated channel is composed of resistances from the boundary and the vegetation stems. Boundary resistance consists of sidewall resistance, and grain and bed form resistances on a mobile bed surface. Grain resistance is the friction due to bed surface roughness, and it is a function of bed roughness height, proportional to the size of bed sediment. While bed form resistance is a drag force due to flow separation at the lee side of bed form, it is also a function of bed form height. The summation of grain and bed form resistances is the total bed resistance. Zanke et al. [9] compared 14 relations for calculating bed form-related friction in non-vegetated channels, and found the method of Engelund [10] to be the most accurate. In vegetated channels, bed resistance could be significantly smaller than that in a non-vegetated channel at the same flow discharge. Jordanova and James [11] and Kothyari et al. [12] calculated bed resistance by subtracting vegetation resistance from the total flow resistance when processing their laboratory experimental data. The vegetation resistance was determined by using the drag coefficient for a single cylindrical stem but taking into account the effect of other adjacent stems (Equation (5) in [11] and Equation (4) in [12]). A numerical modeling study conducted by

Lopez and Garcia [13] found bed resistance in vegetated channels reduced steadily with the increase in vegetation roughness density, defined as $aH$, where $a$ is the vegetation frontal area per unit volume (m$^{-1}$), and $H$ is flow depth (m). For channels with submerged vegetation having $H/h_v = 3$, where $h_v$ is vegetation height, bed resistance is reduced to just 10% of the bare bed value [14,15].

Yang et al. [5] and Yang and Nepf [6,7] found that the near-bed turbulence kinetic energy in the wake of vegetation elements is more important than bed resistance to quantify the sediment transport in vegetated channels. Wang et al. [16,17] developed formulas to quantify the critical flow velocity for incipient sediment movement in the presence of emergent and submerged vegetation. They found that the vegetation drag has an inherent effect on the initiation of sediment motion in vegetated open channel flow because of its impact on turbulence and mean flow. These studies did not provide empirical relations for estimating bed form resistance for engineering applications.

Jordanova and James [11] and Kothyari et al. [12] correlated the bed load transport rate in vegetated channels with bed resistance using laboratory data. However, the results of Jordanova and James [11] are not generic because only one sediment size, one stem diameter, and one vegetation density were used. Kothyari et al. [12] studied the effect of emergent vegetation on bed load transport. They also observed that the bed load transport rate in vegetated channels is smaller than that in non-vegetated ones. Kothyari et al. [12] modified the original bed load transport equation for a non-vegetated channel by Hashimoto and Hirano [18] to account for vegetation resistance. In the comparison, the bed load transport rate in a vegetated channel is a function of bed resistance and the critical shear stress of bed sediment. Another experimental study by Specht [19] investigated bed load transport in a bare bed channel with emergent vegetation on the banks. Since flow velocity on the vegetated banks is less than that in the channel, a secondary current is generated with the direction towards the channel center at the bottom, but outwards at the free surface, resulting in a scour hole at the bank toe. This secondary flow circulation considerably affects the direction of bed load transport. In this experiment, vegetation on the banks accelerated bed load transport in the main channel and consequently caused scour at the bank toe and then bank failure. Apparently, vegetation on the bed and banks have different effects on bed load transport in the main channel.

Although the most recent studies observed enhanced deposition within a vegetated bed surface, the opposite trend has also been observed [20]. Follett and Nepf [21] observed the erosion and deposition patterns formed in an experimental sand bed around a circular patch of emergent vegetation imitated by rigid cylinders. All of their measurements showed some degrees of scouring within the patch. They attributed that to the higher level of turbulence within the vegetation patch. Sediment scoured from the sparse patch was mostly deposited within one patch diameter downstream of the same patch. Additional deposition occurred further downstream but at the sides of turbulence wake, creating an open bed formation (Figure 1a). For a dense patch, flow experienced greater resistance, and sediment scoured from this patch was carried further downstream before being deposited along the patch centerline. Consequently, a closed bed formation was created (Figure 1b). The density of vegetation is apparently a key factor that influences sediment transport and the erosion/deposition pattern in vegetated channels. At what density the presence of vegetation on the channel bed or banks will reduce bed load transport and induce deposition, or accelerate bed load transport and cause scour, remains unknown at present.

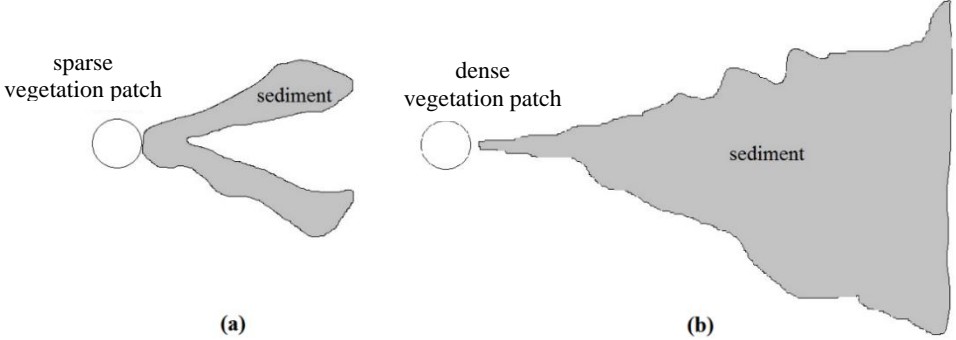

**Figure 1.** Schematic drawing for (**a**) open bed formation, (**b**) closed bed formation.

Therefore, the objective of this study was to conduct a series of laboratory experiments to investigate bed form resistance and bed load transport in a vegetated mobile bed channel. Sediment sizes, flow conditions, and vegetation densities were varied in the experiments. Vegetation, PVC pipes, was mounted on a perforated board (bed surface) covered with 10 cm of sediment. The focus was to find out how vegetation density affects bed load transport and bed form resistance. Empirical relations for calculating bed form resistance and bed load transport rate, respectively, were proposed using this and other experimental data. Coefficients in these relations were optimized for the maximum Nash–Sutcliffe coefficient (NSE) by using the DSM.

## 2. Experimental Setup

### 2.1. Flume Setup

A set of 18 experimental runs were conducted in an open channel flume at the Department of Civil and Architectural Engineering and Mechanics, University of Arizona. The flume was 0.6 m wide and 12.2 m long with a flat bed, and glass and stainless steel sidewalls. A large water tank was used to provide water to the flume. A valve installed at the pipe inlet controlled the flow rate. A sharp crested rectangular weir located at the end of the flume was used to measure flow rate ($Q$). The vegetation stems were simulated by emergent PVC rods of 16 mm outside diameter. The stems were inserted into holes drilled into a 1.5 cm thick and 4.8 m long coated wood board, as shown in Figure 2. Although the rigid cylinders do not have the same flexibility as the natural vegetation, they can replicate the impacts of vegetation stems on flow and sediment in laboratory experimental flumes. The PVC rods have many sizes and are easy to mount into bed surface, and have been commonly used in vegetated channel research [11,12,22,23]. Because of this, the results from this research are suitable to vegetation with sturdy stems and low flexibility.

The stems were arranged in three different configurations of regular staggered grids (Figure 3). In this study, the vegetation concentration, $\phi$, is defined as the fraction of bed area occupied by the vegetation stems = $N\pi d^2/4$, where $N$ is the number of stems per unit bed area, and $d$ is the outsider diameter of stem. The vegetation concentrations for these three configurations are 0.033, 0.014, and 0.005, respectively.

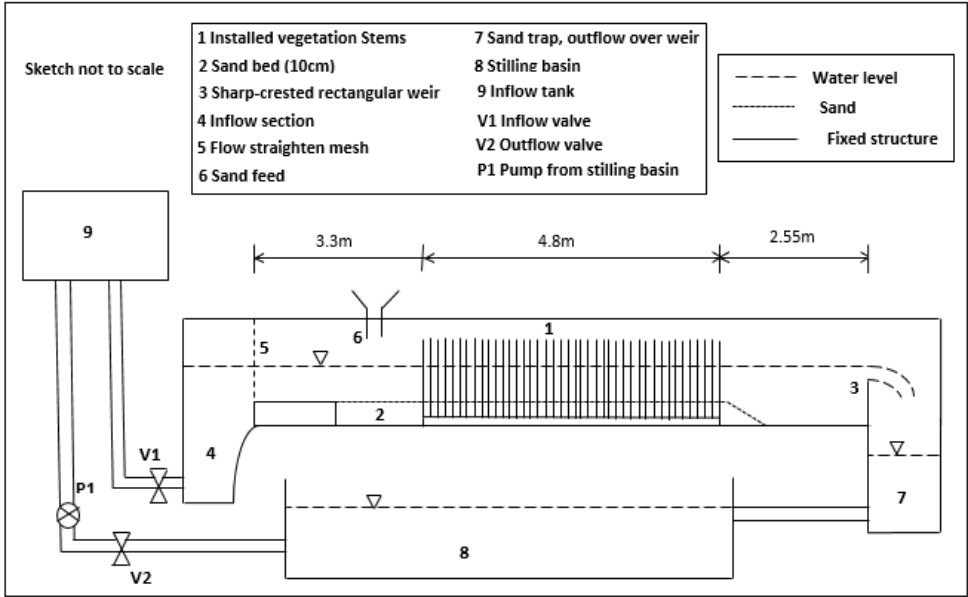

**Figure 2.** Flume setup.

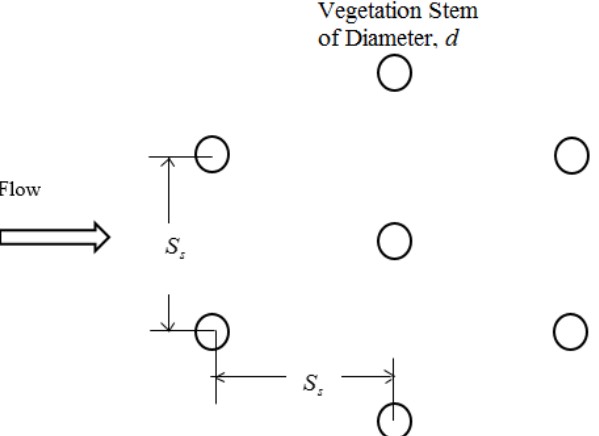

**Figure 3.** Vegetation stems arrangement.

Two groups of weakly non-uniformly-sized sediment with mean sizes $d_{50}$ = 0.45 mm and 1.6 mm, were used. The standard deviations of sediment mixtures defined as $\sigma_g = (d_{84}/d_{16})^{0.5}$, were 1.47 and 1.35 for sediment sizes of 0.45 mm and 1.6 mm, respectively. Sediment mixtures with the value of $\sigma_g$ less than 1.6 are considered as weakly non-uniform [24]. The density of sediment, $\rho_s$, is equal to 2650 kg/m³. Before each experimental run, sediment was saturated and placed evenly on bed surface at a depth of 10 cm (Figure 2).

### 2.2. Water and Bed Slopes

Flow depth was measured at an interval of 0.45 m along the vegetated reach using a fine-scaled ruler of 1 mm accuracy. When the measured flow depths were nearly constant along the measurement reach (~4.8 m), the flow was nearly quasi-steady and quasi-uniform. Each experimental run was typically about 5–6 h. After each experiment, bed elevations were measured by a laser level (accuracy 1 mm) at an interval of 0.45 m in the streamwise and 0.1 m in the transverse directions within the vegetated reach. Bed elevation at each cross-section was the average of all the measured elevations in the transverse direc-

tion. The longitudinal bed slope was obtained by fitting bed elevations at each cross-section with a straight line. The water surface elevations at each cross-section were calculated as the summation of water depth and bed elevation. Then, the friction (energy) slope was calculated by fitting water surface elevations with a straight line. It is noticed that the water surface slope changes at the entrance and exit of the vegetation section, but these measured points were not used for curve fitting of surface slope. For all the experimental runs, the correlations from the curve fitting of bed and friction slopes calculation were greater than or equal to 0.84. Results showed the friction slopes were nearly equal to the measured bed slopes for all the experiments.

### 2.3. Bed Load Transport Rate

During each experimental run, sediment was supplied at the flume entrance to supplement the sediment being washed out of the flume, and in order to keep approximately steady state flow condition. Bed load transport rate, $q_b$, was measured by a bed load sampler at the end of vegetated reach. The sampler nozzle is 4.5 cm high and 25 cm wide, and mounted on a rod. The collection bag has a mesh size of 0.2 mm. The sampling time interval ranged from 1.0 to 8.0 min. The bed load samples were dried and weighted. Bed load transport rate was first calculated as weight per unit time, and then converted to the volume per unit time per unit channel width (m²/s). All the measurements were taken after flow had reached steady state and are summarized in Table 1.

**Table 1.** Experimental runs.

| Run | $d$ (mm) | $d_{50}$ (mm) | Stem Spacing $(S_s)$ (mm) | $N$ | $\phi$ | $S$ (%) | $H$ (cm) | $Q \times 10^3$ (m³/s) | $q_b \times 10^6$ (m²/s) |
|-----|------|------|------|---------|-------|-------|-------|-------|-------|
| 1 | 16 | 0.45 | 78 | 164.366 | 0.033 | 0.805 | 11.4 | 11.34 | 0.95 |
| 2 | 16 | 0.45 | 78 | 164.366 | 0.033 | 1.14 | 15.3 | 18.44 | 2.77 |
| 3 | 16 | 0.45 | 78 | 164.366 | 0.033 | 1.705 | 17.3 | 25.18 | 7.68 |
| 4 | 16 | 1.6 | 78 | 164.366 | 0.033 | 1.4 | 15.1 | 18.87 | 0.51 |
| 5 | 16 | 1.6 | 78 | 164.366 | 0.033 | 1.55 | 15.9 | 21.95 | 3.02 |
| 6 | 16 | 1.6 | 78 | 164.366 | 0.033 | 1.78 | 16.8 | 25.18 | 6.17 |
| 7 | 16 | 0.45 | 120 | 69.4444 | 0.014 | 0.62 | 8.8 | 9.92 | 0.38 |
| 8 | 16 | 0.45 | 120 | 69.4444 | 0.014 | 0.77 | 11.52 | 16.36 | 2.02 |
| 9 | 16 | 0.45 | 120 | 69.4444 | 0.014 | 1.05 | 15.3 | 25.66 | 4.99 |
| 10 | 16 | 1.6 | 120 | 69.4444 | 0.014 | 0.94 | 13.87 | 23.32 | 1.47 |
| 11 | 16 | 1.6 | 120 | 69.4444 | 0.014 | 1.08 | 15.29 | 29.05 | 2.57 |
| 12 | 16 | 1.6 | 120 | 69.4444 | 0.014 | 1.3 | 16.36 | 34.67 | 12.65 |
| 13 | 16 | 0.45 | 200 | 25 | 0.005 | 0.51 | 11.4 | 21.50 | 3.53 |
| 14 | 16 | 0.45 | 200 | 25 | 0.005 | 0.62 | 12.2 | 25.66 | 4.14 |
| 15 | 16 | 0.45 | 200 | 25 | 0.005 | 0.74 | 13.8 | 31.57 | 8.19 |
| 16 | 16 | 1.6 | 200 | 25 | 0.005 | 0.54 | 13.5 | 29.05 | 1.58 |
| 17 | 16 | 1.6 | 200 | 25 | 0.005 | 0.65 | 14.5 | 34.67 | 3.82 |
| 18 | 16 | 1.6 | 200 | 25 | 0.005 | 0.85 | 15.9 | 40.61 | 5.74 |

### 2.4. Bed Surface Elevation

Microsoft Kinect [25] was used to capture a depth image of bed surface for determining bed elevations in the scoured vegetated reach. Microsoft Kinect was initially designed for gaming, but it has many other applications [25–28]. It is composed of RGB-D (Red Green Blue + Depth) sensors. These sensors can produce an RGB visible light image and a depth-coded image from the structured infrared light. The bed elevations are captured by the depth sensor with an accuracy of 640 × 480 pixels. The basic principle of depth sensor is to emit an infrared light pattern and calculate depth from the light reflection at different positions [29]. This generates a depth-coded image that consists of dots with

known coordinates and depths. Kinect is supported in MATLAB (version 2013a), and its data can be acquired using the MATLAB Image Acquisition Toolbox.

## 3. Data Processing

### 3.1. Grain Resistance

Nepf [20] summarized five methods to calculate bed resistance in vegetated channel by using: (i) the spatial averaged viscous stress on bed surface determined by the velocity gradient within the laminar sublayer; (ii) the near-bed turbulent kinetic energy; (iii) the near-bed Reynolds stress; (iv) the difference between total flow and vegetation resistances; and (v) an alternative approach based on the ratio of mean velocity in vegetated layer and the stem diameter. The first method can be used to calculate the grain resistance, and it needs the measurement of velocity profile within the laminar sublayer. This type of measurement has uncertainties due to the turbulence generated from the interaction between flow, vegetation, and bed sediment. The second and third methods are not appropriate for calculating the grain resistance because they include turbulence generated from bed forms and vegetation stems. The fourth method is feasible for calculating the total bed and grain resistances on mobile and fixed beds, respectively. The fifth method is used for calculating the grain resistance, and its applicable range is $aH \geq 0.3$. Unfortunately, this method cannot be used in this study because some of our data and the data obtained in the literature are not within this range. This study employed the fourth method to calculate the total bed resistance.

Recently, Le Bouteiller and Venditti [30] evaluated several methods to calculate the grain resistance in vegetated channel. Among these methods, the inversion of bed load transport formula and the method by Einstein and Banks [31] appeared to yield the most accurate values of grain resistance. Based on the logarithmic velocity distribution, Einstein and Banks [31] and Le Bouteiller and Venditti [30] found the grain resistance can be calculated as:

$$H_s = \frac{U^2}{gS}\left[\frac{1}{\kappa}\cdot\ln\left(\frac{11\,H_s}{k_{sg}}\right)\right]^{-2} \tag{1}$$

$$\tau_g = \rho g\, H_s\, S \tag{2}$$

where $\tau_g$ is grain resistance (N/m²), $\rho$ is water density (kg/m³), $H_s$ is the equilibrium flow depth (m), $S$ is bed slope, $\kappa$ is von Karman constant (= 0.41), $k_{sg}$ is grain roughness height (= $2.5d_{50}$), and $U$ is mean flow velocity (m/s) [= $Q/(BH)$], where $B$ is channel width (m). According to the above two equations, Equation (1) is used to calculate the equilibrium flow depth, and Equation (2) is used to calculate the grain resistance, $\tau_g$.

### 3.2. Sidewall Resistance

The sidewall resistances on the glass, $\tau_{w1}$, and stainless steel, $\tau_{w2}$, were calculated using $\tau_{w1\,or\,w2} = (\rho V_v^2 f_{w1\,or\,w2})/8$, where $V_v$ is mean pore velocity though the vegetation [$V_v = Q/(BH(1\text{-}\phi))$] [12,22]; $f_{w1}$ and $f_{w2}$ are the Darcy–Weisbach friction coefficient for the glass and stainless steel sidewalls, respectively. These coefficients can be obtained by using the Colebrook equation:

$$\frac{1}{\sqrt{f_{w1\,or\,w2}}} = -2\cdot\log[\,\frac{\frac{k_{sw1\,or\,sw2}}{4r}}{3.7} + \frac{2.51}{R_e\cdot\sqrt{f_{w1\,or\,w2}}}\,] \tag{3}$$

where $k_{sw1}$ and $k_{sw2}$ are the roughness heights for glass and stainless steel, respectively. For glass, it is nearly zero, and for stainless steel, $4.5 \times 10^{-5}$ m. $R_e$ is flow Reynolds number defined as $R_e = 4r\cdot V_v/\upsilon$, where $r$ is the total hydraulic radius (m) (see Equation (10)), and $\upsilon$ is the kinematic viscosity of water (m²/s).

*3.3. Bed Form Resistance*

According to the fourth method specified in Nepf [20], in steady uniform flow with vegetation, the downslope gravity force components are equal to the total flow resistance including total bed, sidewalls, and vegetation resistances. The mathematical equation for the force balance is:

$$\gamma \forall S = (\tau_g + \tau_{bf})A_{bed} + F_D + \tau_{w1}A_{w1} + \tau_{w2}A_{w2} \tag{4}$$

where $\gamma$ is the specific weight of water (N/m³), $\forall$ is the volume of water (m³) [ $\forall = A_{bed} \cdot H$ ] in which $A_{bed}$ is bed surface area (m²), $\tau_{bf}$ is bed form resistance (N/m²), $A_{w1}$ and $A_{w2}$ are the areas of the glass and the stainless steel sidewall surface, respectively (m²), and $F_D$ is the vegetation drag force (N). For a reach of unit length and width, *B*, the bed surface area is $A_{bed} = B(1 - \phi)$ and the glass and stainless steel sidewall surface areas are $A_{w1} = A_{w2} = H$.

The drag force acting on vegetation stems in Equation (4), $F_D$, can be calculated as:

$$F_D = \frac{1}{2}\rho C_D NBd HV_v^2 \tag{5}$$

where $C_D$ is the vegetation drag coefficient. From Equations (4) and (5), by knowing *H*, *S*, $\tau_g$, $\tau_{w1}$, $\tau_{w2}$, *B*, *d*, $\phi$, and *Q*, the coefficient of vegetation drag must also be known in order to find bed form resistance ($\tau_{bf}$). Cheng [23] developed an approach to calculate the vegetation drag coefficient for a cylinder located in arrays of emergent cylinders using the pseudo-fluid model. An analogy was made between the cylinder-induced drag in an open channel flow with that induced by the cylinder settling in a stationary fluid. The drag coefficient can be calculated as follows:

$$C_D = \frac{1+S}{1-\phi}C_D' \tag{6}$$

where $C_D'$ is the drag coefficient for the pseudo-fluid model, and defined by:

$$C_D' = 11R_e'^{-0.75} + 0.9\left[1 - e^{\left(-\frac{1000}{R_e'}\right)}\right] + 1.2\left[1 - e^{\left\{-\left(\frac{R_e'}{4500}\right)^{0.7}\right\}}\right] \text{ for } 1 \le R_e' \le 10^5 \tag{7}$$

where $R_e'$ is Reynolds number for the pseudo-fluid model, and can be calculated by:

$$R_e' = \frac{1+S}{1+80\phi}\frac{V_v d}{\upsilon}\frac{r_{vm}}{r_v}\; ; \; r_v = (\pi/4)(1-\phi)d/\phi \tag{8}$$

where $r_v$ is the vegetation-related hydraulic radius for vegetated flows without sidewall and bed effect (m); $r_{vm}$ is the modified vegetation-related hydraulic radius (m), and calculated by taking the effect of bed and sidewall resistances into consideration [22]. The term ($r_{vm}/r_v$) was included to account for the corrections to bed and sidewall resistances due to the presence of vegetation [23]. The calculation of $r_{vm}$ was proposed by Cheng and Nguyen [22] as:

$$r_{vm} = r_v\left[1 - \left(\frac{f_b}{H} + \frac{f_{w1} + f_{w2}}{B(1-\phi)}\right)\frac{r}{f}\right] \tag{9}$$

where *f*, $f_{w1}$, and $f_{w2}$ are the total Darcy–Weisbach friction coefficient, and that for the glass and stainless steel sidewalls, respectively; $f_b$ is the bed friction coefficient, and equal to the summation of grain and bed form friction coefficients ($f_b = f_g + f_{bf}$). The *r* value was calculated as follows:

$$r = \left(\frac{1}{H} + \frac{1}{r_v} + \frac{1}{0.5B(1-\phi)}\right)^{-1} \tag{10}$$

In order to calculate $\tau_{bf}$ from measured flow properties (e.g., $H$, $S$) and vegetation parameters ($\phi$, $d$), and channel geometry ($B$) using Equation (4), the trial and error method is adopted because vegetation drag coefficient $C_D$ is a function of $r_{vm}$, which depends on bed form resistance itself. The detailed calculation procedure is as follows:

Step #1: Calculate the grain resistance using Equations (1) and (2), and convert it into the Darcy–Weisbach grain friction coefficient using $f_g = (8\tau_g/(\rho V_v^2))$.

Step #2: Calculate the vegetation-related hydraulic radius, $r_v$, using $r_v = (\pi/4)(1-\phi)d/\phi$, the total hydraulic radius, $r$, using Equation (10), and $f$ value using $f = (8\,g\,r\,S/V_v^2)$.

Step #3: Calculate the glass and stainless steel Darcy–Weisbach friction coefficients using the Colebrook equation, Equation (3), and then calculate sidewall resistances using $\tau_{w1\,or\,w2} = (\rho V_v^2 \cdot f_{w1\,or\,w2})/8$.

Step #4: Assume the Darcy–Weisbach bed friction coefficient, $f_b$. For the first trial, this guess must be greater than $f_g$. Then, perform the following steps to recalculate bed friction coefficient:

1.  Calculate the modified vegetation-related hydraulic radius, $r_{vm}$, using Equation (9).
2.  Calculate Reynolds number for the pseudo-fluid model, $R_e'$, using Equation (8).
3.  Calculate the drag coefficient for the pseudo-fluid model, $C_D{'}$, using Equation (7).
4.  Calculate the vegetation drag coefficient, $C_D$, using Equation (6).
5.  Calculate the vegetation drag force, $F_D$, using Equation (5).
6.  Calculate the bed form resistance, $\tau_{bf}$, using Equation (4).
7.  Calculate the Darcy–Weisbach bed form friction coefficient using $f_{bf} = 8\tau_{bf}/(\rho V_v^2)$.
8.  Recalculate the Darcy–Weisbach bed friction coefficient using $f_b = f_g + f_{bf}$.
9.  Repeat step #4 until the difference between the calculated and the assumed values of $f_b$ is within a desired tolerance.

The calculated $\tau_g$, $\tau_{w1}$, $\tau_{w2}$, $V_v$, $C_D$, and $\tau_{bf}$ values for all the experiments are shown in Table 2. All the experimental flows are subcritical with Froude number ($F_r$) ranging from 0.162 to 0.343. This approach of separating the total bed resistance into grain and bed form resistances was also applied for estimating bed load transport rate in one-dimensional hydrodynamic model [32].

**Table 2.** Experimental runs processing data.

| Run | $\tau_g$ (N/m²) | $\tau_{w1}$ (N/m²) | $\tau_{w2}$ (N/m²) | $V_v$ (cm/s) | $C_D$ | $\tau_{bf}$ (N/m²) | $\Delta Z$ (mm) | $F_r$ |
|---|---|---|---|---|---|---|---|---|
| 1 | 0.334 | 0.079 | 0.081 | 17.15 | 1.18 | 3.264 | 5.5 | 0.162 |
| 2 | 0.484 | 0.107 | 0.110 | 20.78 | 1.15 | 6.279 | 5.4 | 0.170 |
| 3 | 0.712 | 0.148 | 0.153 | 25.09 | 1.12 | 11.513 | 5.6 | 0.193 |
| 4 | 0.884 | 0.114 | 0.118 | 21.54 | 1.16 | 8.708 | 6.1 | 0.177 |
| 5 | 1.037 | 0.136 | 0.140 | 23.80 | 1.13 | 9.204 | 5.2 | 0.191 |
| 6 | 1.209 | 0.156 | 0.162 | 25.84 | 1.12 | 11.029 | 6.5 | 0.201 |
| 7 | 0.360 | 0.096 | 0.099 | 19.06 | 1.12 | 2.944 | 7.7 | 0.205 |
| 8 | 0.528 | 0.139 | 0.143 | 24.00 | 1.04 | 4.220 | 7.3 | 0.226 |
| 9 | 0.731 | 0.180 | 0.186 | 28.35 | 1.01 | 7.905 | 7.8 | 0.231 |
| 10 | 1.089 | 0.183 | 0.190 | 28.42 | 1.00 | 5.324 | 6.3 | 0.244 |
| 11 | 1.339 | 0.225 | 0.234 | 32.12 | 0.97 | 6.138 | 5.7 | 0.262 |
| 12 | 1.646 | 0.271 | 0.283 | 35.82 | 0.96 | 7.770 | 7.0 | 0.283 |
| 13 | 0.691 | 0.224 | 0.233 | 31.60 | 0.95 | 2.759 | 9.2 | 0.299 |
| 14 | 0.853 | 0.270 | 0.282 | 35.23 | 0.94 | 3.607 | 13.0 | 0.322 |

| 15 | 1.012 | 0.308 | 0.323 | 38.32 | 0.93 | 5.069 | 15.0 | 0.329 |
| 16 | 1.245 | 0.277 | 0.289 | 36.05 | 0.92 | 2.527 | 5.4 | 0.313 |
| 17 | 1.524 | 0.332 | 0.347 | 40.05 | 0.92 | 3.260 | 10.3 | 0.336 |
| 18 | 1.818 | 0.369 | 0.387 | 42.78 | 0.92 | 5.844 | 21.7 | 0.343 |

### 3.4. Bed Form Height

After each experimental run, water was slowly drained out of the flume. When bed surface was still wet, vegetation stems were removed carefully to make this region clear for taking images. Microsoft Kinect together with the MATLAB Image Acquisition Toolbox was used to capture a depth image of bed surface at the region indicated above. Using the MATLAB program, we converted the depth-coded image, and stored it as point clouds. Each point in the cloud has $X$, $Y$, and $Z$ values representing its position in space ($X$ and $Y$) as well as its distance or depth ($Z$) from the Kinect depth sensor.

Because bed surface has a longitudinal and a transverse slope, these slopes affect the distance ($Z$-value) from each point on bed surface to the Kinect depth sensor (Figure 4a). In order to remove these effects, hereafter, called bias, another MATLAB program was used for leveling the bed surface (Figure 4b).

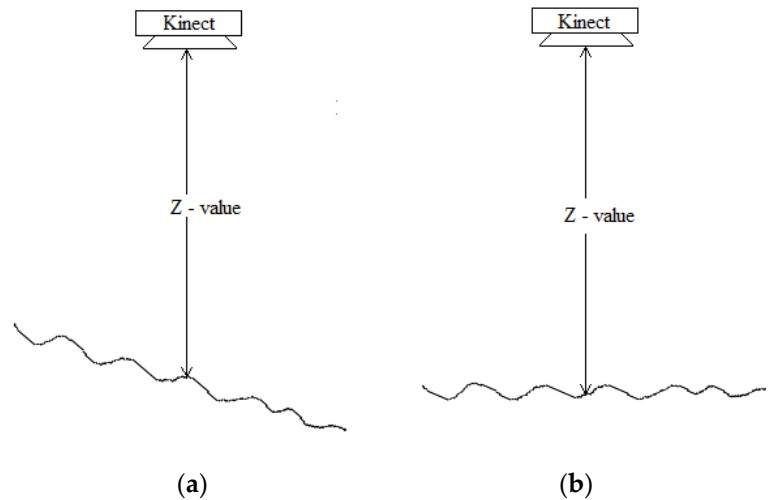

(**a**)                                                      (**b**)

**Figure 4.** Kinect Z-values, (**a**) with bed surface slope, (**b**) leveling bed surface.

The height of bed elevation, $\Delta Z$, shown in Table 2, is defined as the average bed form height, calculated by $\sqrt{\dfrac{\sum\left(Z_i - \overline{Z}\right)^2}{n}}$ , where $Z_i$ is the bed elevation at point $i$; $\overline{Z}$ is the mean bed elevation (m), which is a constant for a horizontal plane; $n$ is the total number of measurement points. Typical bed surface elevation contours, such as run #7 and #18, are shown in Figure 5a,b, respectively.

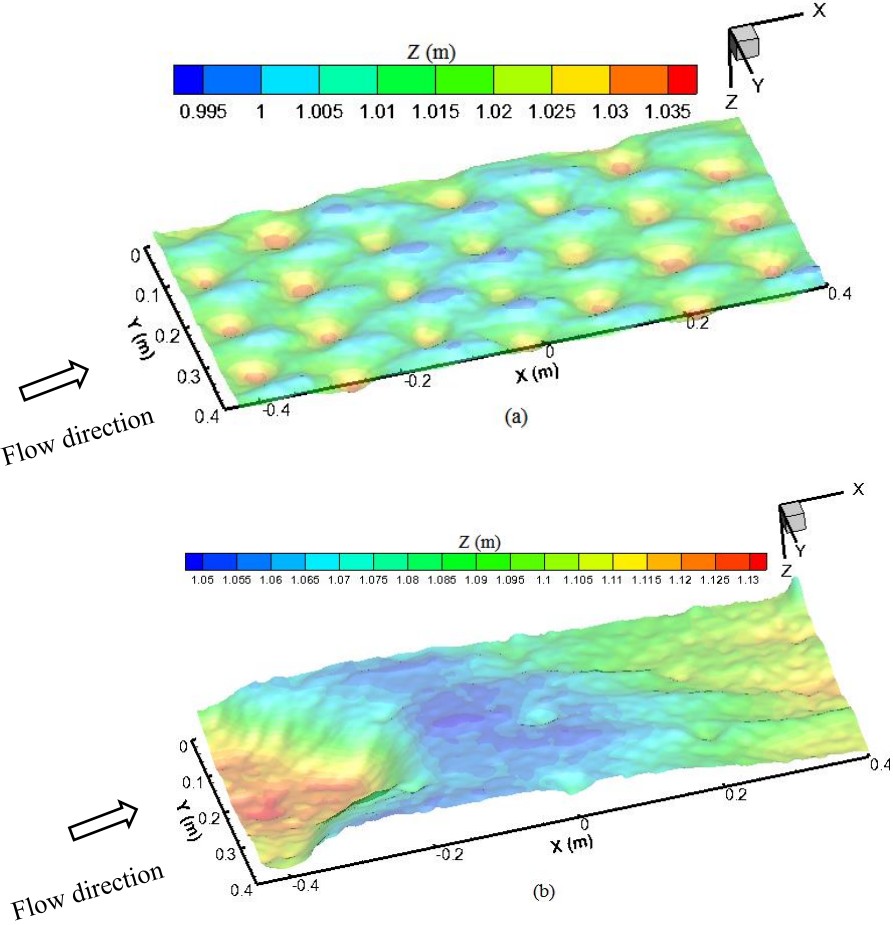

**Figure 5.** Bed surface elevation contours (**a**) run #7 {$\phi$ =0.014}, (**b**) run #18 {$\phi$ =0.005}.

For $\phi$ = 0.033 and 0.014, scour holes were observed around each vegetation stem with a depositional bed form in between (Figure 5a). These bed forms were formed by the high level of turbulence from overlapping wakes and horseshoe vortices generated by each stem. Regardless of flow properties and sediment sizes, the $\Delta Z$-values for $\phi$ = 0.033 were nearly the same (Table 2), and the same trend was noticed for $\phi$ = 0.014. This means that bed form height is highly correlated with vegetation concentration but not flow and sediment properties. The mean values of bed form height, $\Delta Z_{avg}$, for $\phi$ = 0.014 and 0.033 are equal to 7.0 and 5.7 mm, respectively. This shows that bed form height was slightly decreased with the increasing of vegetation concentration.

For $\phi$ = 0.005, because of the formation of sand dunes, the $\Delta Z_{avg}$ value is equal to 12 mm larger than that for other $\phi$ values. For $d_{50}$ = 0.45 mm, as shown in Table 2, the $\Delta Z$ value is slightly increased because smaller sized sand dunes were observed. When $d_{50}$=1.6 mm, as shown in Figure 5b and Table 2, the $\Delta Z$ values are increased as flow velocity is increased due to the increasing of sand dunes' sizes. This implies that sand dunes were developed through the sparse vegetation as flow velocity increases.

The variation of $\Delta Z_{avg}$ versus $\phi$ (Figure 6) indicated that the $\Delta Z_{avg}$ is decreased rapidly as the $\phi$ value increased from 0.005 to 0.014, and then decreased gradually as the $\phi$ value increased from 0.014 to 0.033. This trend is consistent with the evolution of bed form from sand dunes at low vegetation concentration to fully developed scour holes around each vegetation stem at high concentration.

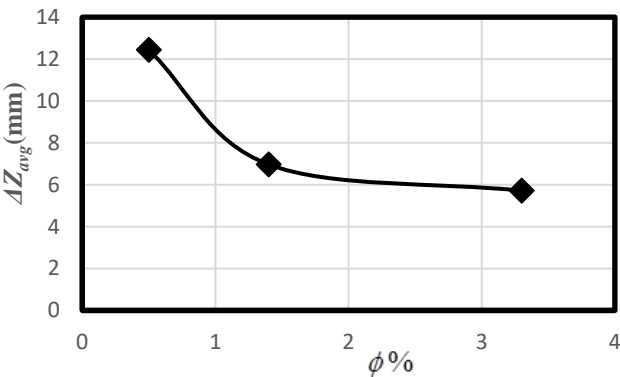

**Figure 6.** Variation of $\Delta Z_{avg}$ versus $\phi$.

## 4. Empirical Relations and Methods

### 4.1. Bed Form Resistance Relation

Bed form resistance is a function of flow, sediment, and vegetation properties, which can be calculated as [30]:

$$\tau_{bf}^* = \frac{C_{bf}}{2}\frac{\rho}{(\rho_s - \rho)gd_{50}}U_{bf}^2\frac{\Delta}{\lambda} \tag{11}$$

in which $C_{bf}$ is bed form drag coefficient for bed form, $\Delta$ is the height of bed form, $\lambda$ is the length of bed form, and $U_{bf}$ is the mean velocity within the height of bed form for bed surface free of vegetation, which is proportional to the vegetation concentration and mean pore velocity as:

$$U_{bf} = \alpha_1\phi^{\beta_1}V_v \tag{12}$$

The non-dimensional bed form resistance can be written as:

$$\tau_{bf}^* = \frac{C_{bf}}{2}\frac{\rho}{(\rho_s - \rho)gd_{50}}U_{bf}^2\frac{\Delta}{\lambda} \tag{13}$$

Based on Sturm [33], the height and length of dunes in non-vegetated channels can be calculated by:

$$\frac{\Delta}{H} = \begin{cases} 0.11(\frac{d_{50}}{H})^{0.3}(1-e^{-0.5T})(25-T) & \text{for } T < 25 \\ 0 & \text{for } T \geq 25 \end{cases} \tag{14}$$

$$\lambda = \begin{cases} 7.3H, \text{for } T < 25 \\ 0 \text{ for } T \geq 25 \end{cases} \tag{15}$$

in which $T$ is bed mobility factor defined as $T = \tau_g/\tau_c - 1$. Bed form is at low energy regime (i.e., ripple, dune) when $T < 25$, and high energy regime (i.e., dynamic flat bed) when $T \geq 25$. For ripples and dunes at $T < 25$, the ratio of bed form height and length is:

$$\frac{\Delta}{\lambda} = \frac{0.11}{7.3}(\frac{d_{50}}{H})^{0.3}(1-e^{-0.5T})(25-T) \tag{16}$$

For vegetated channel, the ratio of bed form height and length is proportional to the ones observed in non-vegetated channel as:

$$\frac{\Delta}{\lambda} = \alpha_2\phi^{\beta_2}\frac{0.11}{7.3}(\frac{d_{50}}{H})^{0.3}(1-e^{-0.5T})(25-T) \tag{17}$$

where $\alpha_2$ and $\beta_2$ are coefficients for taking account of the vegetation impacts. Substituting Equations (12) and (17) for Equation (13), the relation for non-dimensional bed form resistance is written as:

$$\tau_{bf}^* = \frac{C_{bf}}{2} \alpha_1^2 \alpha_2 \frac{0.11}{7.3} \frac{V_v^2}{(G_s-1)gd_{50}} (\frac{d_{50}}{H})^{0.3} \phi^{2\beta_1+\beta_2} (1-e^{-0.5T})(25-T) \tag{18}$$

where $\tau_{bf}^*$ is the non-dimensional bed form resistance, defined as $\tau_{bf}^* = \frac{\tau_{bf}}{[(\rho_s-\rho)gd_{50}]}$. This definition of non-dimensional bed form shear stress is recommended in Zanke and Roland [34]. In Equation (18), set $C_1 = \frac{C_{bf}}{2}\alpha_1^2\alpha_2\frac{0.11}{7.3}\frac{1}{(G_s-1)}$, $C_2 = 2\beta_1 + \beta_2$, and $F_r = \frac{V_v}{\sqrt{gH}}$, the simplified expression of Equation (18) is:

$$\tau_{bf}^* = C_1 Fr^2 (\frac{H}{d_{50}})^{0.7} \phi^{C_2} (1-e^{-0.5T})(25-T) \tag{19}$$

Equation (19) is derived based on semi-empirical relations of bed form properties in non-vegetated channels, and cannot be considered as an accurate description of bed form resistance in vegetated channels. Nevertheless, Equation (19) outlines the non-dimensional variables for bed form resistance calculation, which are Froude number, the ratio of flow depth and sediment diameter, vegetation concentration, and mobility parameter.

### 4.2. Bed Load Transport Relation

Similarly, bed load transport rate is the product of bed load particle velocity and the thickness of bed load layer [34–37]. The thickness of bed load layer is defined as the equivalent bed thickness assuming the bed load layer consists of only bed load particles. The real thickness of mobile bed load layer is greater than this value due to the porosity of bed material. Shim and Duan [37] conducted a series of laboratory experiments using high-speed camera to measure bed load particle velocity, and found the spatial and temporal averaged bed load particle velocity in non-vegetated channel can be expressed as:

$$u_b = \sqrt{\frac{(G_s-1)gd_{50}}{G_s+C_M} (\omega_1 \frac{\tau_g^*}{\tau_c^*} + \omega_2)\tau_g^*} \tag{20}$$

where $u_b$ is bed load particle velocity in non-vegetated channel, $\tau_g^*$ is non-dimensional value of grain resistance $[= \tau_g / \{(\rho_s-\rho)g\,d_{50}\}]$, $G_s$ is the specific gravity of the sediment and equals 2.65, $C_M$ is the added mass coefficient for sediment particles in water, which has a theoretical value of 0.5 for spherical particles [38], $\omega_1$ and $\omega_2$ are coefficients that correlate bed load saltation length to non-dimensional bed shear stress originated in Equations (12) and (13) in Shim and Duan [37]. Bed load velocity in vegetated channels is assumed to be proportional to the one in non-vegetated channels as:

$$u_{b-veg} = \phi^{\alpha_3} u_b \tag{21}$$

where $u_{b-veg}$ is bed load particle velocity in vegetated channel. Equation (21) applies the exponential of vegetation concentration to scale the bed load velocity obtained for non-vegetated channel. Then, bed load transport rate in vegetated channel can be written as:

$$q_b = u_{b-veg}\xi_{b-veg} = \xi_{b-veg}\phi^{\alpha_3} \sqrt{\frac{(G_s-1)gd_{50}}{G_s+C_M} (\omega_1 \frac{\tau_g^*}{\tau_c^*} + \omega_2)\tau_g^*} \tag{22}$$

where $\zeta_{b-veg}$ is bed load layer thickness in vegetated channel. The bed load layer thickness in non-vegetated channel can be estimated as [34]:

$$\frac{\zeta_b}{d_{50}} = \frac{\tau_g^* - \eta\tau_c^*}{\eta\tan\phi_d} \tag{23}$$

where $\eta$ reflects the bed slope angle in the flow direction, which equals 1.0 for mild sloped channels, $\phi_d$ is the dynamic friction angle. For mild sloped channel bed similar to the experimental conditions cited in this paper, $\eta$=1.0 is used in Equation (23). In additions the non-dimensional grain shear replaced the non-dimensional bed shear stress in the original equation [34]. However, Equation (23) is applicable to non-vegetated channel, incorporating the effect of vegetation roots in bed load layer, we assume the thickness of bed load in mild sloped vegetated channel is analogous to Equation (23) as:

$$\frac{\zeta_{b-veg}}{d_{50}} = \phi^{\alpha_4} \frac{(\tau_g^* - \tau_c^*)^{D_1}}{\tan\phi_d} \tag{24}$$

where $\alpha_4$ and $D_1$ are coefficients that differentiate the calculation of bed load thickness in vegetated and non-vegetated channel. Then, Equation (22) can be written as:

$$q_b = \frac{(\tau_g^* - \tau_c^*)^{D_1}}{\tan\phi_d} d_{50} \phi^{\alpha_3 + \alpha_4} \sqrt{\frac{(G_s - 1)gd_{50}}{G_s + C_M}(\omega_1 \frac{\tau_g^*}{\tau_c^*} + \omega_2)\tau_g^*} \tag{25}$$

In Equation (25), set $D_2 = \alpha_3 + \alpha_4$. The non-dimensional bed load transport rate is calculated as:

$$q_b^* = \frac{q_b}{\sqrt{(Gs-1)gd_{50}^3}} = \sqrt{\frac{1}{Gs + C_M}} \frac{(\tau_g^* - \tau_c^*)^{D_1}}{\tan\phi_d} \phi^{D_2} \sqrt{(\omega_1 \frac{\tau_g^*}{\tau_c^*} + \omega_2)\tau_g^*} \tag{26}$$

where $q_b^*$ *is* the non-dimensional bed load transport $(q_b^*)$ [= $q_b / \sqrt{(G_s - 1)gd_{50}^3}$ ]. In Equation (26), $C_M$ = 0.5 for spherical particles. The dynamic friction angle is assumed to be $35°$ for medium size sand. Shim and Duna [37] found $\omega_1 = 26.3$ and $\omega_2 = 34.6$ from experimental data. In this study, the non-dimensional critical shear stress is 0.034. Unfortunately, there is no measurements of saltation particle length in vegetated channel. Therefore, we assumed these constants are valid for vegetated channels. Then, Equation (26) can be further simplified as:

$$q_b^* = 0.8(\tau_g^* - \tau_c^*)^{D_1} \phi^{D_2} \sqrt{(26.3\frac{\tau_g^*}{\tau_c^*} + 34.6)\tau_g^*} \tag{27}$$

where $D_1$ and $D_2$ are empirical coefficients that need to be determined by observed data.

## 5. Downhill Simplex Method to Determine the Coefficients

In order to find the coefficients in Equations (19) and (27), data from this study as well as data from Jordanova and James [11] and Kothyari et al. [12] (Table 3) were used for optimization. In Table 3, the total bed resistance, $\tau_b$, is the sum of grain ($\tau_g$) and bed form ($\tau_{bf}$) resistances, which were calculated in Jordanova and James [11] and Kothyari et al. [12]. By knowing the grain resistance ($\tau_g$) (Equation (2)) and the total bed resistance ($\tau_b$), bed form resistance ($\tau_{bf}$) is calculated as the difference between $\tau_b$ and $\tau_g$. The non-dimensional bed form resistance, $\tau_{bf}^*$, can be calculated based on the experimental measurements. Similarly, the non-dimensional bed load transport rate, $q_b^*$, can also be determined from experimental data. The non-dimensional variables based on experimental measurements were used to optimize the coefficients in Equations (19) and (27).

**Table 3.** Ranges of literature test data considered.

| Investigator | $d$ (mm) | $d_{50}$ (mm) | $\phi$ | $S$ (%) | $H$ (cm) | $V_v$ (cm/s) | $\tau_b$ (N/m²) | $F_r$ | $q_b \times 10^6$ (m²/s) |
|---|---|---|---|---|---|---|---|---|---|

| Kothyari et al. [12] | 2.0 to 5.0 | 0.55 to 5.9 | 0.002 to 0.012 | 1.7 to 20.8 | 2.78 to 6.08 | 33.8 to 94.9 | 1.70 to 59.43 | 0.44 to 1.78 | 0.5 to 8121 |
|---|---|---|---|---|---|---|---|---|---|
| Jordanova & James [11] | 5.0 | 0.45 | 0.0314 | 1.18 to 1.84 | 2.05 to 11.1 | 15.5 to 18.5 | 0.51 to 1.32 | 0.16 to 0.37 | 1.89 to 6.94 |

The downhill simplex method (DSM) [39] was applied to all the data mentioned above. The DSM is a commonly applied optimization technique for determining the minimum or maximum value of an objective function in a multidirectional space [40–43]. It is especially effective in non-linear problems when the derivatives are unknown. In this study, the Nash–Sutcliffe efficiency (NSE) coefficient was maximized and the corresponding coefficient of determination ($R^2$) was calculated to find the best match between the predictions from Equations (19) and (27) and the experimental data measured (observed). The NSE and $R^2$ values were calculated using the following two equations:

$$\text{NSE} = 1 - \frac{\sum(P - O)^2}{\sum(O - \overline{O})^2} \qquad \text{NSE} < 1 \tag{28}$$

$$R^2 = \frac{(m\sum P \cdot O - \sum P \cdot \sum O)^2}{[m\sum P^2 - (\sum P)^2] \cdot [m\sum O^2 - (\sum O)^2]} \tag{29}$$

where P and O are predicted from Equation (19) or (27) and observed values in Tables 1–3; m is the total number of data. The DSM was employed to find optimal coefficient combinations that correspond to functional minimums (or maximums). The optimization starts with an initial simplex, which represents a randomly generated coefficient set. The functional minimum (or maximum) in the coefficient space can be found by transforming the simplex according to the functional values at the vertexes and moving the simplex in the downhill direction until the functional value converges to its minimum (or maximum) value [39].

### 5.1. Optimal Coefficient Set in Bed Form Resistance Relation

To find the optimum coefficient set for $\tau_{bf}^*$ relation (Equation (19)), the DSM optimization was run in MATLAB platform (R2018a, MathWorks, Inc., Natick, MA, USA), by maximizing the NSE values (or minimizing 1.0−NSE values). Since Equation (19) is only for the ripple- and dune-type bed form, we removed the 11 pieces of data in Kothyari et al. [12] because $T \geq 25$. Seven hundred iterations were run to find the optimum values of the coefficients $C_1 = 205.156$ and $C_2 = 2.484$. Unfortunately, the maximum NSE value was only 0.034 and $R^2$ was 0.117. This indicates Equation (19) is not an appropriate function for bed form resistance. In order to improve the NSE value, we modified the constants in Equation (19) and added calibration coefficients while maintaining the original variables. The modified equation can be written as:

$$\tau_{bf}^* = C_1 Fr^{C_2} \left(\frac{H}{d_{50}}\right)^{C_3} \phi^{C_4} (1 - e^{-0.5T})(25 - T)^{C_5} \tag{30}$$

We used five coefficients (Equation (30)) instead of two (Equation (19)) to rerun the DSM analysis, and conducted 700 iterations until the coefficients converged to constants (Figure 7). This figure shows the influence of the coefficient values' variation on (1.0−NSE) values throughout the DSM iterations. The optimum coefficient values yielded the minimum 1.0−NSE value of 0.109, corresponding to the maximum NSE value of 0.891. The coefficients were $C_1 = 0.238$, $C_2 = 0.519$, $C_3 = 1.533$, $C_4 = 0.638$, and $C_5 = -1.034$. The corresponding $R^2$ value for this optimum coefficient set was 0.890 as shown in the scatter plot (Figure 8). These values of the NSE and $R^2$ show that the predicted Equation (30) and the observed non-dimensional bed form resistances are correlated well for all the datasets.

The positive exponential for vegetation concentration indicates bed form resistance increases with vegetation concentration.

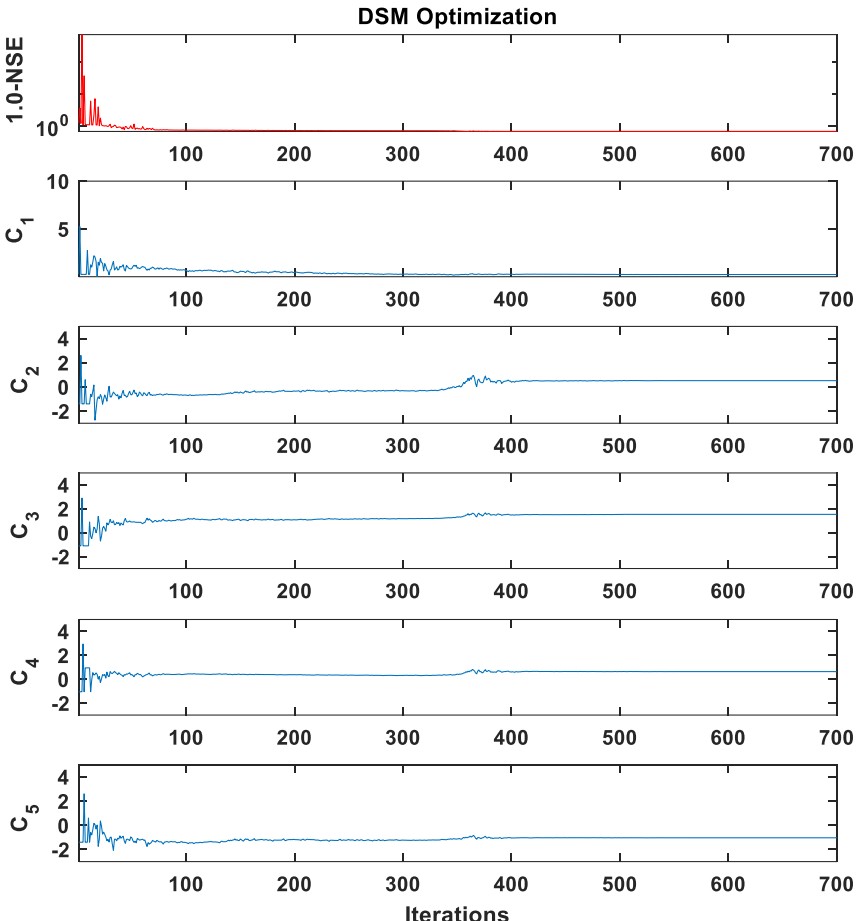

**Figure 7.** DSM optimization results for Equation (30) by minimizing 1.0−NSE (or maximizing NSE).

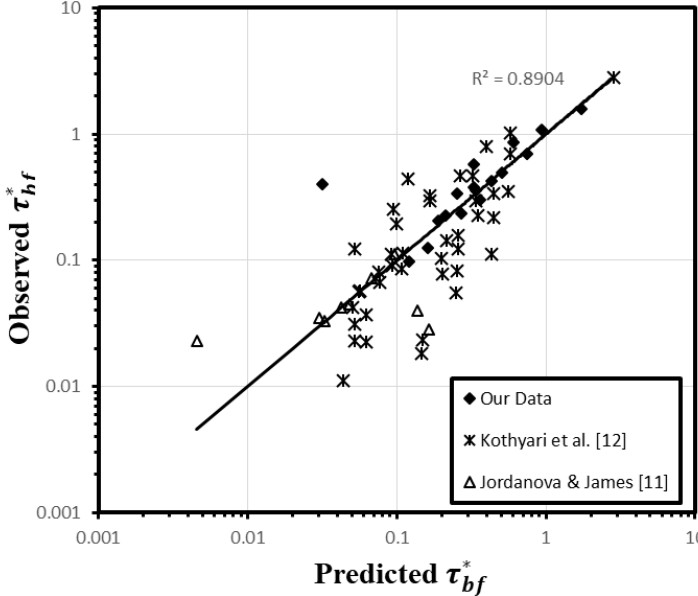

**Figure 8.** Scatter plot for predicted values of $\tau_{bf}^*$ (Equation (30)) and observed ones based on optimum parameter set.

### 5.2. Optimal Coefficient Set for Bed Load Transport Relation

To find the optimum coefficient set for $q_b^*$ relation (Equation (27)), the DSM optimization ran 200 iterations by maximizing the NSE values (or minimizing 1.0−NSE values). The influence of these coefficients ($D_1$ and $D_2$) on 1.0−NSE values throughout the DSM iterations is shown in Figure 9. Apparently, both coefficients converge to constants at the maximum NSE value. The optimum coefficient values for the minimum 1.0−NSE value of 0.019 (or the maximum NSE value of 0.981) are $D_1 = 1.919$ and $D_2 = -0.168$. The corresponding $R^2$ value for the above optimum coefficient set is 0.981 as shown in the scatter plot (Figure 10). The resultant equation of the bed load transport rate with the optimal coefficients is as below:

$$q_b^* = 0.80467(\tau_g^* - \tau_c^*)^{1.919} \phi^{-0.168} \sqrt{(26.3\frac{\tau_g^*}{\tau_c^*} + 34.6)\tau_g^*} \tag{31}$$

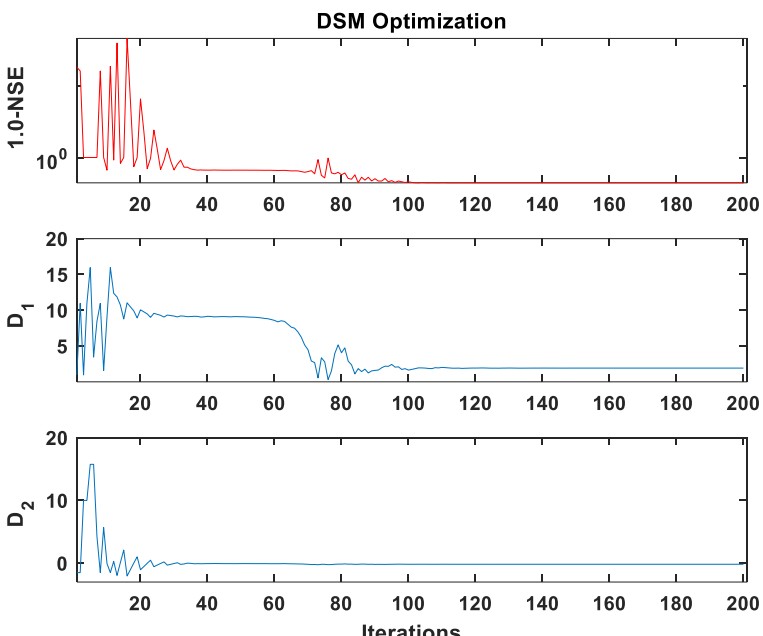

**Figure 9.** DSM optimization results for Equation (27) by minimizing 1.0−NSE (or maximizing NSE).

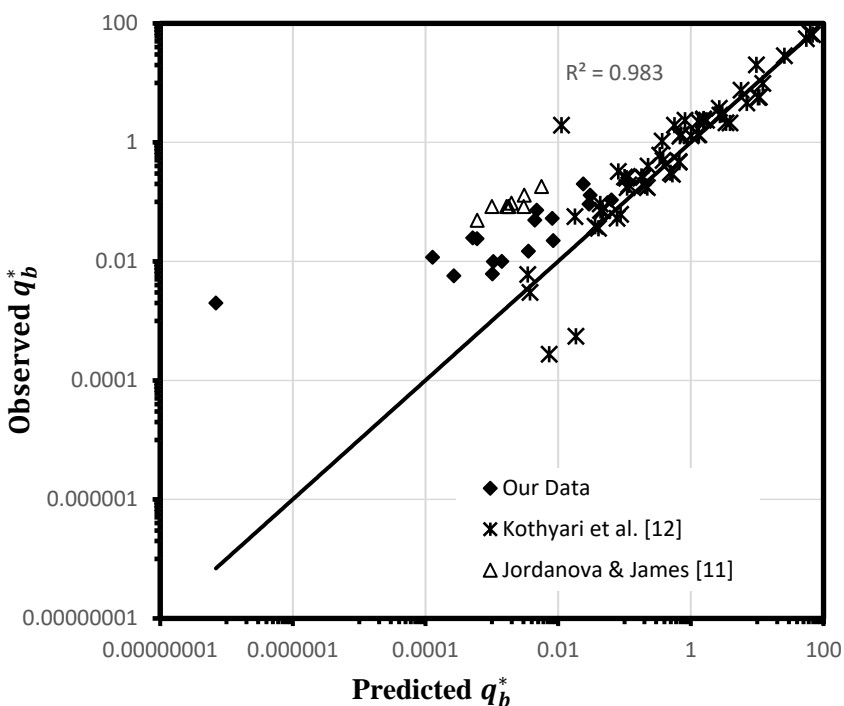

**Figure 10.** Scatter plot for predicted and observed values of $q_b^*$ based on optimum parameter set.

The coefficient $D_2 = -0.168$ in Equation (31) is negative, meaning the bed load transport rate is reducing with vegetation concentration. If the total bed resistance remains as a constant, the increase in vegetation concentration will increase bed form resistance (Equation (30)) so that grain resistance will be reduced. Consequently, the bed load transport rate reduces with vegetation concentration as seen in Equation (31). However, $D_2 = -0.168$, which is close to zero. Therefore, to quantify the influence of the vegetation concentration on the prediction of the bed load transport rate, Equation (31) is reevaluated using DSM optimization without the $\phi$ parameter. The values of NSE and $R^2$ are 0.653 and 0.985, respectively. Although the maximum values of $R^2$ with and without the incorporation of the vegetation concentration are approximately the same, the maximum NSE value with the incorporation of the vegetation concentration (NSE = 0.981) is greater than the one without the vegetation parameter (NSE = 0.653). This means that the vegetation concentration has a moderate effect on the prediction of bed load transport (Equation (31).

## 6. Discussion

Vegetation drag force is dependent on the drag coefficient, $C_D$, which varies with flow Reynold number, and approaches a constant for a single cylinder in fully turbulent flow. In this study, $C_D$ is required in each experiment for quantifying the drag force induced by vegetation stems. In order to determine the resistance on vegetation stems (Equation (5)), the grain resistance and the sidewall resistance were calculated first by Equations (2) and (3), respectively. Second, the total bed shear stress, consisting of grain resistance and bed form resistance, were assumed in Step #4. By knowing the grain and sidewall resistances, the vegetation drag coefficient $C_D$ was calculated in Step #4.1–4.4. Third, the vegetation resistance and bed form resistance were calculated in Step #4.5 and Step #4.6, respectively. Fourth, we converted bed form resistance to the Darcy–Weisbach resistance coefficient (Step #4.7). In Step #4.8-4.9, we recalculated the bed resistance. This bed resistance must be equal to the assumed bed resistance; otherwise, the assumed bed resistance was adjusted until it converged. This procedure calculates the bed form resistance after knowing

the grain resistance and vegetation drag resistance. The $C_D$ value is dependent on the vegetation stem Reynolds number ( $Re_d = V_v d / \nu$ ) (Figure 11), flow Reynolds number ( $Re_H = UH / \nu$ ) (Figure 12), and vegetation concentration ($\phi$) (Figure 13).

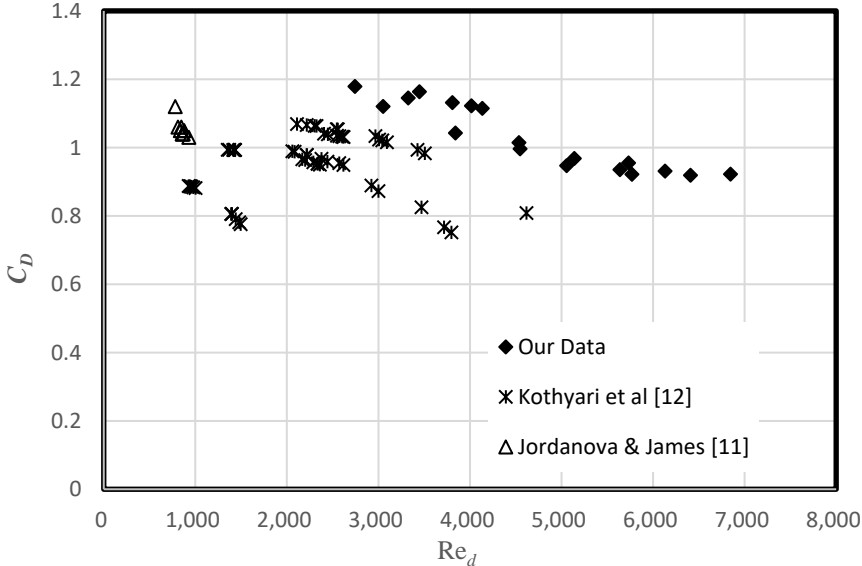

**Figure 11.** $C_D$ versus vegetation stem Reynolds number.

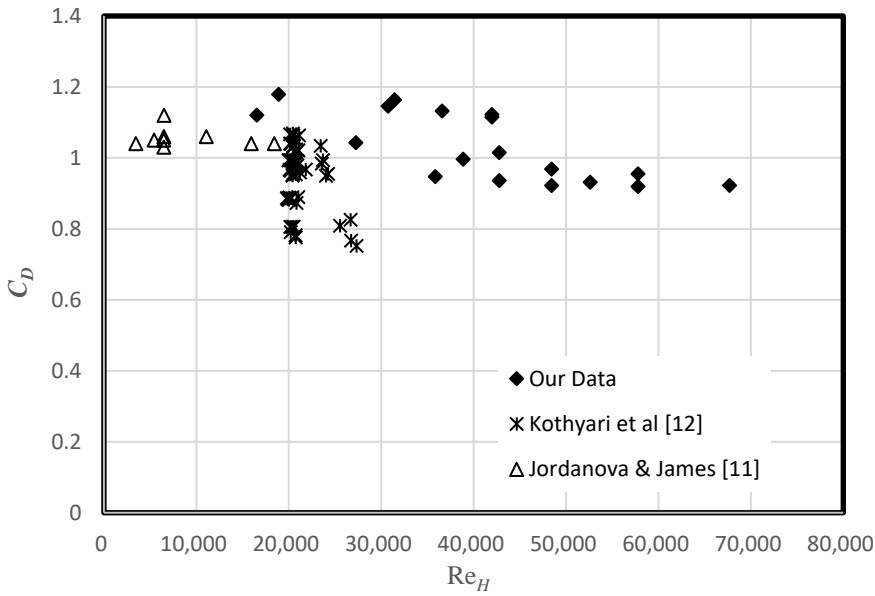

**Figure 12.** $C_D$ versus flow Reynolds number.

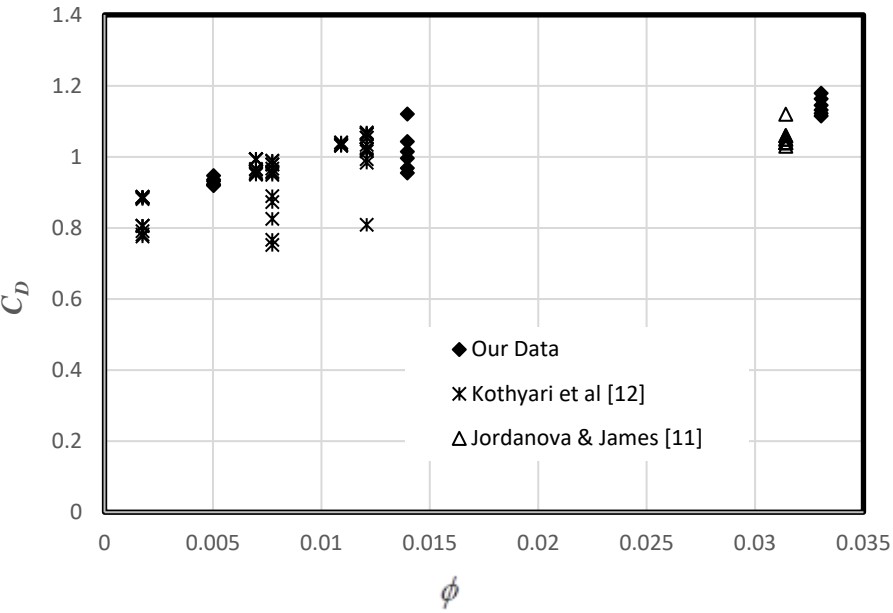

**Figure 13.** $C_D$ versus vegetation concentration.

These figures showed that $C_D$ ranges from 0.75 to 1.18 for vegetation concentration from 0.002 to 0.033, vegetation stem Reynolds number ($780 < \mathrm{Re}_d < 6844$), and flow Reynolds number ($3444 < \mathrm{Re}_H < 67675$). The stem Reynolds number indicates the turbulence strength near the stems, while the flow Reynolds number measures the turbulence in the main flow. As we have seen, flow is fully turbulent, with the minimum Reynolds number of 3444. Therefore, the $C_D$ coefficient varies within a narrow range from 0.75 to 1.18 based on the trial–error calculation. Figures 11 and 12 show that $C_D$ reduces as the Reynolds number increases, which is consistent with other studies [20,23]. Figure 13 shows that $C_D$ increases with vegetation concentration due to the impact of overlapped vorticity structures.

In addition, traditional bed load formulas are used for the estimation of the bed load transport rate in non-vegetated channels [44], such as the formulas of Einstein [45], Meyer-Peter and Muller [46], and Bagnold [47] for uniform bed material, and Parker [48], Duan and Scott [49], and Wilcock and Crowe [50] for mixed-sized bed material. These bed load transport formulas cannot be applied directly to predict the bed load transport rate in a vegetated channel for two major reasons: (1) grain resistance for bed load transport is much smaller in a vegetated channel than in a non-vegetated channel due to the vegetation drag force; (2) the higher the vegetation concentration, the lower the bed load transport rate. If directly using the formulas for a non-vegetated channel, the bed load transport rate will be artificially over-predicted, and, consequently, computational models will predict erosion in vegetated channels, contradictory to the reality. As vegetation grows in channels, grain resistance will reduce and the bed load transport rate will reduce. As a result, deposition will be likely to occur in vegetated channels. Therefore, Equation (31) is a novel contribution for estimating bed load transport in vegetated channels.

## 7. Conclusions

A series of laboratory experiments were conducted in an open channel flume to study bed form resistance and bed load transport in a vegetated mobile bed channel. The logarithmic velocity distribution was used in this study to calculate the grain resistance [30,31]. In a vegetated mobile bed, bed form is a series of scour holes around vegetation stems

overtopped on ripples or dunes. The height of bed form depends on vegetation concentration, which determines whether the ripple/dune or scour holes dominate on the bed surface. A new iterative method was derived to calculate bed form resistance. For sparsely vegetated flows, bed form height decreases rapidly as the vegetation concentration is increased, and then decreases gradually upon high vegetation concentration. In our experiments, sand dunes started to appear when the vegetation was sparse, and their sizes increased with flow velocity. Empirical relations were derived to predict bed form resistance and the bed load transport rate. The DSM optimization was applied to find the coefficient sets for each relation by maximizing the NSE values and finding the corresponding $R^2$ values. The results showed that vegetation concentration has moderate impacts on bed form resistance and bed load transport. As vegetation concentration increases, bed form resistance will increase, while the bed load transport rate will reduce. This explains that a high-density vegetated channel blocks bed load transport to downstream reaches. Nevertheless, these conclusions were drawn from the limited laboratory experimental data, and require additional data of sediment transport in a densely vegetated channel and field to verify their applicability.

**Author Contributions:** Conceptualization, J.G.D. and K.A.-A.; formal analysis, J.G.D. and K.A.-A.; funding acquisition, J.G.D.; methodology, J.G.D. and K.A.-A.; investigation, K.A.-A.; software, K.A.-A.; visualization, K.A.-A.; writing—original draft, K.A.-A.; project administration, J.G.D.; resources, J.G.D.; supervision, J.G.D.; validation, K.A.-A. and J.G.D.; writing—review and editing, J.G.D. and K.A.-A. All authors have read and agreed to the published version of the manuscript.

**Funding:** NSF Award EAR-0846523 and Pima County Regional Flood Control District.

**Institutional Review Board Statement:** Not applicable.

**Informed Consent Statement:** Not applicable.

**Data Availability Statement:** Experimental data from this study will be available upon request. Other data can be found in relevant literature [11,12].

**Acknowledgments:** Special thanks to Hoshin V. Gupta for providing the DSM program.

**Conflicts of Interest:** The authors declare no conflict of interest.

## Notation

The following symbols are used in this paper:

| | |
|---|---|
| $a$ | vegetation frontal area per unit volume (m$^{-1}$); |
| $aH$ | vegetation roughness density (-); |
| $A_{bed}$ | bed surface area (m$^2$); |
| $B$ | channel width (m); |
| $C_D$ | drag coefficient for a cylindrical emergent stem; |
| $C_D'$ | drag coefficient for the pseudo-fluid model; |
| $C_{bf}$ | bed form drag coefficient (-); |
| $d$ | vegetation stem diameter (mm); |
| $d_{16}, d_{84}$ | sizes for which 16% and 84% of the sediment are finer than $d_{16}$ and $d_{84}$, respectively (mm); |
| $d_{50}$ | median sediment size (mm); |
| $F_D$ | vegetation drag force (N); |
| $Fr$ | Froude number (-); |
| $g$ | gravity acceleration (m/s$^2$); |
| $G_s$ | specific gravity of the sediment (-); |
| $H$ | flow depth (m); |
| $H_s$ | equilibrium flow depth (m); |
| m | total number of data; |
| $N$ | number of stems per unit bead area (m$^{-2}$); |

| NSE | Nash–Sutcliffe efficiency coefficient (-); |
| $n$ | total number of bed elevation points; |
| $O$ | observed value of non-dimensional bed form resistance or non-dimensional bed load transport rate; |
| $P$ | predicted value of non-dimensional bed form resistance or non-dimensional bed load transport rate; |
| $Q$ | flow rate (m³/s); |
| $q_b$ | bed load transport rate (m²/s); |
| $q_b^*$ | non-dimensional bed load transport; |
| $R^2$ | coefficient of determination (-); |
| $Re^{'}$ | Reynolds number for the pseudo-fluid model (-); |
| $Re_H$ | flow Reynolds number (-); |
| $Re_d$ | vegetation stem Reynolds number (-); |
| $r$ | total hydraulic radius (m); |
| $r_v$ and $r_{vm}$ | vegetation-related, and modified vegetation-related hydraulic radii respectively (m); |
| $S$ | bed slope (-); |
| $S_s$ | vegetation stem spacing (mm); |
| $T$ | bed mobility factor (-); |
| $U$ | mean flow velocity (m/s); |
| $U_{bf}$ | mean velocity within the height of bed form (m/s); |
| $u_b$ | bed load particle velocity in non-vegetated channel (m/s); |
| $u_{b\text{-}veg}$ | bed load particle velocity in vegetated channel (m/s); |
| $V_v$ | mean pore velocity (m/s); |
| $X$, $Y$, and $Z$ | position of the bed points (points clouds) in space and distance (depth) (m); |
| $Z_i$ | bed elevation at any point i (m); |
| $\overline{Z}$ | mean bed elevation of original bed surface (m); |
| $\Delta$ | height of bed form; |
| $\Delta Z$ | bed form height (mm); |
| $\Delta Z_{avg}$ | average of the bed form height for each $\phi$ value (mm); |
| $\zeta_b$ | bed load layer thickness in non-vegetated channel; |
| $\zeta_{b\text{-}veg}$ | bed load layer thickness in vegetated channel; |
| $f$ | total Darcy–Weisbach friction coefficient (-); |
| $f_b$ | bed Darcy–Weisbach friction coefficient= $f_g + f_{bf}$ (-); |
| $f_g$, $f_{bf}$ | grain and bed form Darcy–Weisbach friction coefficients, respectively (-); |
| $\phi$ | vegetation concentration (-); |
| $\phi_d$ | dynamic friction angle; |
| $\gamma$ | specific weight of water (N/m³); |
| $\lambda$ | length of bed form; |
| $\mu$ | water dynamic viscosity (N · s/m²); |
| $\nu$ | kinematic viscosity of water (m²/s); |
| $k_{sg}$, $k_{sw1}$, $k_{sw2}$ | grain, glass, and stainless steel roughness heights (m) |
| $\kappa$ | von Karman constant = 0.41 (-); |
| $\rho$, $\rho_s$ | water and sediment density, respectively (kg//m³); |
| $\sigma_g$ | standard deviation of sediment mixture; |
| $\tau_b$ | total bed resistance (N/m²) = $\tau_g + \tau_{bf}$; |
| $\tau_g$, $\tau_{bf}$ | grain and bed form resistances, respectively (N/m²); |
| $\tau_{w1}$, $\tau_{w2}$ | glass and stainless steel sidewall resistances, respectively (N/m²); |

| $\tau_{bf}^*$, $\tau_g^*$, $\tau_c^*$ | non-dimensional bed form, grain, and critical resistances, respectively; |
| $\forall$ | volume of water (m³); |
| $\beta_1$, $\beta_2$, $\alpha_1$, $\alpha_2$, $\alpha_3$, $\alpha_4$, $C_1$, $C_2$, $C_3$, $C_4$, $C_5$, $\eta$, $D_1$, $D_2$, $\omega_1$, $\omega_2$ | coefficients. |

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
