# Peer review of "On Bed Form Resistance and Bed Load Transport in Vegetated Channels"

_water, doi:10.3390/w14233794_

Round 1

Reviewer 1 Report

The manuscript presents investigations on the effect of vegetation on the Roughness and Bed Load Transport in channels with movable bed and vegetation. The topic is of interest, but the paper needs to be revised in parts.

Title:

The title should be specified to "On Bed Form Resistance and Bed Load Transport in Vegetated Channels"

Abstract line 13:

Please indicate that bedforms mean ripples and dunes.

Introduction lines 34-35 and chapter 3.3:

With respect to bedform roughness, the recenty pulished paper ("Roughness Effects of Subaquaeous Ripples and Dunes" (MDPI WATER 2022 (14)) should be considered.

The lines 217-218 and in the chapter "Conclusions" should be clarified. The sentence

Based on Einstein and Banks (1950), Le Bouteiller and Venditti (2015) calculated the grain resistance as:

should be expressed like

Based on the logarithmic velocity distribution law resulting from the work of Prandtl and v.Karman, Einstein and Banks (1950), Le Bouteiller and Venditti (2015) calculated the grain resistance under the assumption of hydraulical fully rough flow as:

In line 78 and elsewhere in the text

it is mentioned that the vegetation is imitated by rigid cylinders. Here a statement about the flexibility and the possible oscillation behavior of the model vegetation and the natural vegetation is needed.

Figure 1:

In the figure and its caption, the vegetation patches should be referred to as "loose" and "dense" vegetation.

Line 147:

The sediment used is a mixture, so it is better to call it "weakly non-uniform" rather than uniform.

Figure 4:

Please link the figure to Fig. 5 by indicating corresponding profiles in Fig. 5.

Figure 5:

Add the direction of flow.

Line 368:

In the equation for tau*_bf, replace gamma with g.

Chapter 4.2

Regarding q*_b, please consider the article "Sediment Bed-Load Transport: A Standardized Notation" (MDPI geosciences, 2020 (10)).

Reviewer 2 Report

This study uses laboratory flume experiments with dowels to study the impacts of vegetation on bedform resistance and bedload transport. The research topic is very important to the hydrology and geomorphology community. The study used many existing semi-empirical equations and also proposed new scaling equations. However, the applicable conditions and uncertainties of each equation have not been clearly discussed. The form of the proposed new equation has not been justified. Uncertainty and error analyses are missing. Many experimental details, such as the duration of the experiments and the equilibrium conditions, are missing. Uncertainty analysis is critical for experimental studies but is lacking in this study.

My detailed comments are as follows.

Section 2.2: There is no flow velocity measurements. How long were the experiments? Have the flow reached equilibrium?

Section 2.3: it is not clear how the sediment was supplied into the flume, especially when the sediment input has to equal to output at equilibrium.

Equations 1 (a) and (b): It is not clear how vegetation would impact these equations. Equation 1(b) is definitely going to change when there is vegetation.

Equation 5 and 6: I think the vegetation drag coefficients CD always has a large uncertainty and varies with environmental conditions. It is not clear at what Reynolds number range these equations apply and what the constraints are.

Lines 282-302: The steps to calculate bed stress and other stress involve many empirical equations and coefficients. I am not sure how accurate the final results would be. Especially when the vegetation drag is dominant, or when Tau_g + tau_bf is much smaller than vegetation drag, a small uncertainty in CD would generate huge uncertainties in Tau_g and Tau_bf. I think there should be analysis and justification of uncertainty.

Equation 11: I understand that we can express non-dimensional shear stress as a function of other non-dimensional variables, but why the equation looks like this (the exponential form) is not clear. There need to be explanation on how this form of equation is derived.

Same for equation 13.

Equations 16-17: There are so many significant digits in these equations. Are they necessary?

Round 2

Reviewer 1 Report

Chapter 4.2
The focus of the reviewer's comment is not on the comparability of formulas.  Since in many practical cases the effect of vegetation can only be qualified by comparison with calculations without vegetation, a reference to literature on the current status of sediment transport calculation options without vegetation is helpful to the reader here.

In general
Some literature is cited by name and number, some by number only. Please standardize.

Reviewer 2 Report

The authors have improved the manuscript by adding more details. However, I still think could not under agree with many of the equations used in the manuscript. 

For example, in areas with vegetation, the flow will not follow logarithmic distribution, so I do not think it is appropriate to use log fitting to find grain resistance. 

In addition, there are still no error bars for parameters that have large uncertainty such as CD. The authors used three significant digits, e.g. 1.18, to estimate CD. I think this is confusing, because in general CD can have uncertainty up to 20 to 50 percent or large. 

Round 3

Reviewer 2 Report

I think the authors have addressed all my concerns. Despite the difference in our science perspective, I think the evidence the authors provided support their conclusion.